# Molecular Mechanisms of Cytotoxicity of NCX4040, the Non-Steroidal Anti-Inflammatory NO-Donor, in Human Ovarian Cancer Cells

**DOI:** 10.3390/ijms23158611

**Published:** 2022-08-03

**Authors:** Birandra K. Sinha, Erik J. Tokar, Carl D. Bortner

**Affiliations:** 1Mechanistic Toxicology Branch, Division of the National Toxicology Program, National Institute of Environmental Health Sciences, Research Triangle Park, NC 27709, USA; erik.tokar@nih.gov; 2Laboratory of Signal Transduction, Division of Intramural Research, National Institute of Environmental Health Sciences, Research Triangle Park, NC 27709, USA; botner@niehs.nih.gov

**Keywords:** NCX4040, nitric oxide, peroxynitrite, reactive oxygen species, DNA damage

## Abstract

NCX4040, the non-steroidal anti-inflammatory-NO donor, is cytotoxic to several human tumors, including ovarian tumor cells. We have found that NCX4040 is also cytotoxic against both OVCAR-8 and its adriamycin resistant (NCI/ADR-RES) tumor cell lines. Here, we have examined mechanism(s) for the cytotoxicity of NCX4040 in OVCAR-8 and NCI/ADR-RES cell lines. We found that NCX4040 induced significant apoptosis in both cell lines. Furthermore, NCX4040 treatment caused significant depletion of cellular glutathione, causing oxidative stress due to the formation of reactive oxygen/nitrogen species (ROS/RNS). Significantly more ROS/RNS were detected in OVCAR-8 cells than in NCI/ADR-RES cells which may have resulted from increased activities of SOD, glutathione peroxidase and transferases expressed in NCI/ADR-RES cells. NCX4040 treatment resulted in the formation of double-strand DNA breaks in both cells; however, more of these DNA breaks were detected in OVCAR-8 cells. RT-PCR studies indicated that NCX4040-induced DNA damage was not repaired as efficiently in NCI/ADR-RES cells as in OVCAR-8 cells which may lead to a differential cell death. Pretreatment of OVCAR-8 cells with N-acetylcysteine (NAC) significantly decreased cytotoxicity of NCX4040 in OVCAR-8 cells; however, NAC had no effects on NCX4040 cytotoxicity in NCI/ADR-RES cells. In contrast, FeTPPS, a peroxynitrite scavenger, completely blocked NCX4040-induced cell death in both cells, suggesting that NCX4040-induced cell death could be mediated by peroxynitrite formed from NCX4040 following cellular metabolism.

## 1. Introduction

Nitric oxide (^●^NO) is a short-lived free radical molecule that has been suggested to act as a cellular messenger. ^●^NO is also implicated in many important physiological functions [1]. Activities of ^●^NO in cells have been shown to result from formation of reactive metabolites, N_2_O_3_, and –OONO. Furthermore, it has been reported that these metabolites rapidly react with sulfhydryl groups of cellular proteins, resulting in modulations of both their activity and stability [2]. Various human tumors express high activity of iNOS, the inducible form of nitric oxide synthase [3,4]. 

We have reported that ^●^NO, or its metabolites, inhibits ATPase activity of ABC transporters in human MDR tumor cells [5]. Because ^●^NO and/or its metabolites are cytotoxic to tumor cells [6], our laboratory has been involved in identifying tumor-specific cytotoxic nitric oxide donors that can be utilized for the reversal of drug resistance caused by the overexpression of ABC transporters in tumor cells [5,7,8]. We have shown that JS-K, a highly cytotoxic NO-donor, was effective in inducing reversal of ADR resistance in a P-gp-overexpressing human tumor cell line [7]. However, it failed to reverse drug resistance in a BCRP-overexpressing tumor cell line. Recently we have shown that NCX4040, the non-steroidal anti-inflammatory NO-donor, effectively modulated drug resistance in both P-gp- and BCRP-overexpressing human cell lines [8]. NCX4040 is a nitro derivative of aspirin and has been shown to release ^●^NO following hydrolysis by tumor esterases [9] as confirmed by EPR [10]. NCX4040 has been reported to be significantly cytotoxic to various tumor cells both in vitro and in vivo [11]. Furthermore, it has been shown to reverse cis-platinum resistance in tumor cells [10]. Our previous studies have shown that NCX4040 is cytotoxic to both human ovarian OVCAR-8 and its P-gp-expressing NCI/ADR-RES cell lines [8]. Ovarian carcinoma, which has an extremely high rate of mortality, is one of the leading causes of gynecological cancers in women. Treatment of ovarian cancer in the clinic consists of platinum drugs in combinations with taxanes and/or adriamycin. Both taxanes and adriamycin are substrates for P-gp, and therefore, are actively removed from tumors that over express ATP-dependent transporters in the clinic [12]. 

A significant amount of research has now been carried out to understand the mechanism of actions of NCX4040; however, it is not completely clear at this time. NCX4040 has been reported to generate ROS, causing depletion of glutathione, inducing oxidative stress in tumor cells, and leading to apoptosis [9]. Others have proposed the formation of a quinone methide which reacts rapidly with glutathione, causing depletion of glutathione and cell death [13,14,15]. The known and proposed mechanisms of NCX4040 action are summarized in Figure 1. The common theme appears to be the depletion of cellular glutathione following formation of either ROS or a quinone methide. NCX4040 has also been reported to cause DNA damage, resulting either from the generation of ROS or a quinone methide. 

Because NCX4040 was cytotoxic to both OVCAR-8 and OVCAR-8-derived ADR resistant (NCI/ADR-RES) cell lines, we have now examined the mechanisms of action of NCX4040 cytotoxicity in these cell lines. We chose these cell lines to examine the mechanism of action of NCX4040 because of the following reasons: First, NCI/ADR-RES cells are extremely resistant to free radical generating ADR, and this resistance resulted from the high expressions of SOD, catalase and glutathione peroxidase compared to the sensitive OVCAR-8 cells [16,17]. Two, NCI/ADR-RES cells also contained higher amounts of glutathione and higher expression of glutathione transferase (GST) compared to the sensitive OVCAR-8 cells [18,19]. The functions of these antioxidant enzymes and GSH/GST have been shown to protect cells from oxidative stress caused by the formation of ROS as well as other reactive electrophiles generated in tumor cells. Thus, it is expected that ROS generated from NCX4040 in NCI/ADR-RES cells would be rapidly detoxified due to the presence of higher GSH, glutathione peroxidase as well as by SOD and catalase. If ROS is responsible for DNA damage and cell death, then NCI/ADR-RES cells would be resistant to cell killing by NCX4040 compared to OVCAR-8 cells. 

Results presented here show that significantly more ROS were produced in OVCAR-8 cells compared to the NCI/ADR-RES cells which resulted in the formation of significantly more DNA double-strand breaks in OVCAR-8 cells than NCI/ADR-RES cells. We also found that NCI/ADR-RES cells were resistant to lower concentrations of NCX4040, indicating ROS played a significant role in cell death. Our studies also showed that while NAC was effective in decreasing NCX4040-induced cells’ death in OVCAR-8 cells, it had no effect in the resistant NCI/ADR-RES cells. In contrast, FeTPPS significantly inhibited cytotoxicity of NCX4040 in both types of cell, suggesting that cytotoxicity may be mediated by peroxynitrite.

## 2. Results

### 2.1. Cytotoxicity Studies with NCX4040 and Topotecan

As shown in Figure 2, NCX4040 is cytotoxic to both OVCAR-8 and NCI/ADR-RES cells. However, at lower doses of NCX4040, NCI/ADR-RES cells showed a limited resistance to cell killing. At higher concentrations, no significant resistance was observed and IC_50_ was very similar to the corresponding OVACAR-8 cells (Figure 2A). While TPT is not usually considered a good substrate for P-gp, some reports have indicated that TPT is a weak substrate for P-gp [20]. Our study (Figure 2B) shows that NCI/ADR-RES cells are resistant to TPT; however, this resistance is significantly small (only 3–4-fold) compared to other substrates of P-gp e.g., adriamycin or etoposide (>100-fold) as found before [18].

### 2.2. Glutathione Depletion by NCX4040 in OVCAR-8 and NCI/ADR-RES Cells 

Previous studies have shown that treatment of tumor cells with NCX4040 leads to depletion of GSH, causing oxidative stress [9]. Results (Figure 3) show that NCX4040 treatment significantly depleted GSH in both cells in a concentration dependent manner. It is interesting to note that more GSH was depleted at higher concentrations of NCX4040 (>10.0 µM) in NCI/ADR-RES cells than in OVCAR-8 cells.

### 2.3. Formation of Reactive Oxygen Species from NCX4040

Because studies have shown that NCX4040 generates ROS in some tumor cells, we used Mitosox to detect these species in both OVCAR-8 and NCI/ADR-RES cells. While use of Mitosox has been questioned for the selective detection of superoxide anion radical [21], it does detect mitochondrial ROS and has been utilized by various investigators [22,23]. Figure 4 clearly shows that a significant amount of ROS was detected in the sensitive OVCAR-8 cells compared to NCI/ADR-RES cells, increasing with time. Lower levels of ROS formation and detection may be due to rapid detoxifications of active species from higher amounts of SOD, catalase, and GPx expressed in the ADR resistant NCI/ADR-RES cells.

### 2.4. Formation and Detection of DNA-Double Strand Breaks by NCX4040

NCX4040 produced significant ROS in these cells and ROS are known to generate highly cytotoxic double-strand breaks (DBS) in DNA. DSB’s formation in OVCAR-8 and NCI/ADR-RES cells were investigated using the H2AX phosphorylation assay. TPT was used as the control as TPT has been shown to induce DSB’s in tumor cells. Since NCI/ADR-RES cells were found to be slightly resistant to TPT (Figure 2B), we used an equitoxic dose of TPT in both cells. Results presented in Figure 5 show that NCX4040 formed significant amounts of DSB’s in both cells; however, more DSB’s were formed in OVCAR-8 cells compared to NCI/ADR-RES cells. This would suggest that the reactive species responsible for the formation of DSB’s in NCI/ADR-RES cells were detoxified due to high expressions of SOD, catalase, and GSH-dependent peroxidase and transferase in NCI/ADR-RES cells.

### 2.5. Induction of Apoptotic Cell Death by NCX4040

Because NCX4040 has been reported to induce apoptosis in tumor cells, we examined the formation of apoptotic cells in OVCAR-8 and NCI/ADR-RES cells using the CaspaTag assay [24]. Our results show that NCX4040 induced significant caspase-mediated apoptosis in these cells (Figure 6). It should be noted that at higher concentrations of NCX4040 (>10.0 µM) significantly more apoptotic cells were detected in NCI-ADR-RES cells compared to OVCAR-8 cells. This was further confirmed by the Annexin V binding assay [25] and is shown in Figure 7.

### 2.6. Effects of NAC and FeTPPS on NCX4040 Cytotoxicity

To further define the mechanism of tumor cell killing by NCX4040, we utilized various inhibitors/quenchers of reactive species-, (e.g., NAC and FeTPPS) and examined their effects on NCX4040 cytotoxicity. While NAC has been used to elucidate roles of ROS as well as peroxynitrite in biological activities [26,27], FeTPPS is shown to be a scavenger of peroxynitrite and is reported to inhibit cellular damage caused by peroxynitrite [28,29]. Data presented in Figure 8 show that NAC (2 mM) was effective in inhibiting NCX4040 cytotoxicity in OVCAR-8 cells. However, it had no effect in modulating NCX4040 cytotoxicity in the resistant NCI/ADR-RES cells. Lower concentrations of NAC were not as effective. In contrast, FeTPPS (5 µM) completely inhibited NCX4040 cytotoxicity in both cells. These observations would then suggest that peroxynitrite may be involved in the cytotoxicity of NCX4040.

### 2.7. RT-PCR Studies in OVCAR-8 and NCI/ADR-RES Cells

RT-PCR studies were used to examine oxidative stress-, glutathione-related, and DNA repair gene expressions following NCX4040 (5.0 µM) treatment in OVCAR-8 and NCI/ADR-RES cells. Our results (Figure 9) shows that NCX4040 treatment resulted in significant modulations of various gene expressions that are related to apoptosis, oxidative stress and DNA repair in both OVCAR-8 and NCI/ADR-RES cells. These genes were examined at an early time (4 h), intermediate (24 h), and later (48 h) time when cells start to show the toxicity of NCX4040.

We found that the anti-apoptotic gene *BCl2* expression was unchanged in both OVCAR-8 and NCI/ADR-RES cells while the pro-apoptotic *BAX* gene expression was rapidly decreased at an early time point in OVCAR-8 cells. In contrast, *BAX* gene was significantly upregulated by NCX4040 in NCI/ADR-RES cells at all time points examined. Heme Oxygenase (*HMOX1* or *OX1*) gene, a marker for oxidative stress, was significantly (4.5-fold and 7.3-fold) elevated in both cell lines at 4 h following NCX4040 treatment and it rapidly decreased at both 24 h and 48 h. Glutathione-related genes, glutathione reductase (*GSR*) and glutathione peroxidase (*GPx*), were not significantly modulated in OVCAR-8 cells by NCX4040; however, these genes were upregulated (2–3-fold) in NCI/ADR-RES cells. DNA Glycosyse1 (*OGG1*), a DNA repair gene responsible for repairing oxidative DNA damage, with 8-Oxoguanine generated in DNA by ^●^OH, decreased slightly at 4 h but it was similar to control values at 24 h and 48 h in OVCAR-8 cells while no significant effects of NCX4040 were observed in NCI/ADR-RES cells. In addition, *RAD51*, a DNA double-strand repair gene, was significantly induced (3–14-fold) at 4 h and 24 h, respectively in OVCAR-8 cells. In contrast, *RAD51* was not significantly modulated by NCX4040 treatment at either 4 h or 24 h in NCI/ADR-RES cells. However, *RAD51* was significantly decreased by NCX4040 at 48 h in both cell types. It should be noted that *GADD45*, a growth arrest and DNA damage response gene, was significantly decreased in NCI/ADR-RES cells while it was slightly induced in OVCAR-8 cells. These observations, taken together, indicate that overall DNA damage (oxidative and global) response genes are induced in OVCAR-8 cells while the DNA damage repair related genes are decreased in NCI/ADR-RES cells. This would suggest that the DNA damage and the resulting double-strand DNA breaks in NCI/ADR-RES cells by NCX4040 persist for a longer time, causing cell death. 

## 3. Discussion

Finding a complete cure of human cancers in the clinic is essential. Significant number of drugs have been synthesized as chemotherapeutic agents without much success as cancer cells are heterogeneous and contain cells that are either inherently resistant or acquire resistance during treatment. Therefore, rationally designed selective anticancer drugs are critically needed. Delivery of high amounts of a cytotoxic species such as ^●^NO to tumor cells may be an important step towards this goal due to two important observations: First: ^●^NO is cytotoxic to tumor cells and second, ^●^NO inhibits ATPase activities of ABC transporters and thus it can overcome MDR. Because the efflux proteins require the ATP/ATPase system to remove drugs from their cellular targets, inhibition of ATPase activities by ^●^NO in drug-resistant tumor cells is essential for the reversal of MDR in the clinic [30]. 

Our laboratory is focused in identifying various nitric oxide donors that are extremely cytotoxic to tumor cells and that can also overcome drug resistance without showing significant toxicity in the clinic. We found that JS-K, a GSH/GST activated NO-donor, was cytotoxic to tumor cells, and could overcome resistance in P-gp-mediated multi-drug resistance tumor cells [7]. Nitric-oxide-donating nonsteroidal anti-inflammatory drugs, including NCX4040, consisting of an NO-donating moiety covalently attached to aspirin, were originally designed as chemo-preventive drugs. NCX4040 was found to be extremely potent in inhibiting colon tumor cells and was shown to be active in preventing colon cancer in animal tumor models with no significant toxicity. NCX4040 is also cytotoxic to various tumors cells both in vitro and in xenograft model systems and has been a subject of extensive research [10,11,13,31,32]. We have shown that NCX4040 is cytotoxic to both OVCAR-8 and its ADR-selected resistant OVCAR-8 (NCI/ADR-RES) cells [8]. Furthermore, NCX4040 significantly decreased drug-resistance mediated by both P-gp and BCRP in human tumor cells. Various investigators have shown that NCX4040 induces oxidative stress, depletes GSH, and leads to DNA damage in tumor cells [9,13]. While NCX4040 has been proposed to induce both the formation of ROS and a quinone-methide (Figure 1), the mechanism(s) of cytotoxicity of NCX4040 in tumor cells is not clear and deciphering the mechanism is important to rationally design better and selective analogs of NCX4040 as anticancer drug.

In order to elucidate the mechanism of NCX4040-induced tumor cell death, we chose human ovarian cell lines, a sensitive and adriamycin-selected multi-drug resistant cell line. NCI/ADR-RES cells have been shown to overexpress P-gp and various antioxidant and phase II enzymes (SOD, Catalase, GSH-based peroxidase, and transferase) compared to sensitive OVCAR-8 cells [16,18]. We believe these cell lines provide excellent means to examine the mechanism of actions of NCX4040 as they are expected to distinguish among different mechanisms of NCX4040 due to increased expression of both antioxidant and phase II enzymes in NCI/ADR-RES cells and thus, will decrease formation of ROS due to enhanced detoxification of these species [17,33]. 

Studies presented here show that NCX4040 formed significant amounts of ROS in both OVCAR-8 cells and its resistant variants; however, decreased amounts of ROS were detected in resistant cells as expected due to enhanced detoxification of ROS. Furthermore, less of DNA double-strand breaks were observed in NCI/ADR-RES cells due to decreased ROS and peroxynitrite (O=N-O-O^−^), formed from superoxide and ^●^NO, or other DNA damaging species formed. ^●^NO (or peroxynitrite) has been shown to induce DSB’s in tumor cells [34]. Peroxynitrite has been reported to diffuse quite far from the site of formation on a cellular scale as it is able to move across cell membranes [35]. It should be noted that less DNA damage was also observed with TPT in the resistant NCI/ADR-RES cells. TPT is a topoisomerase I poison and is considered to kill tumor cells by causing topoisomerase I-dependent DNA double-strand breaks [36], however, TPT also generates free radicals and induces ROS-mediated DNA damage [37]. 

Our studies also show that NAC, a non-specific scavenger of ROS and peroxynitrite, was effective in inhibiting NCX4040 cytotoxicity in OVCAR-8 cells, but it had no effects in the resistant NCI/ADR-RES cells. The reason for this is not clear, however, it is possible that NAC does not enter or is not transported into the resistant cells due to the overexpression of P-gp, resulting in decreased amounts of NAC. Furthermore, we have shown that FeTPPs, a scavenger of peroxynitrite, completely attenuated NCX4040 cytotoxicity in both cells. These observations suggest that NCX4040 may be metabolized to peroxynitrite in these cells, inducing cell death in both OVCAR-8 and NCI/ADR-RES ovarian cells. 

Evaluations of differentially expressed genes following NCX4040 treatment by RT-PCR clearly support that both OVCAR-8 and NCI/ADR-RES cells undergo oxidative stress as *HMOX1*/*OX1*, a biomarker for oxidative stress, was rapidly induced by NCX4040 in both cells. *HMOX1*/*OX1* is induced during oxidative stress by other drugs in tumor cells and mediates antioxidant and cytoprotective effects from ROS-induced cell death [38,39,40]. We found other oxidative stress related genes, *GSR* and *GPx*, were also significantly induced in NCI/ADR-RES cells, indicating increased detoxification of ROS in NCI/ADR-RES cells. The pro-apoptotic *BAX* was significantly induced in NCI/ADR-RES cells compared to OVCAR-8 where *BAX* was rapidly decreased while *BCl2* remained unchanged in both cell lines. This would indicate that apoptosis in these cells is *BAX*-mediated. Our CaspaTag and Annexin V binding studies suggested that NCX4040-induced apoptosis was driven by the activation of caspase 3/7 in both OVCAR-8 and NCI/ADR-RES cells. 

The other interesting observation relates to the effects of NCX4040 on DNA damage-associated genes, *RAD51* and *GADD45*. It should be noted that both OVCAR-8 and NCI/ADR-RES are p53 mutant cell lines and express the p53 with deleted 21-bp and 18-bp within the DNA-binding domain [41,42]. In addition, NCI/ADR-RES cells have been shown to contain arginine instead of proline at codon 72 of the p53 [42]. Using RT-PCR, we were unable to detect wild-type p53 in these cell lines. *RAD51* which catalyzes the core reactions of homologous recombination (HR) DNA double-strand breaks [43,44], was significantly induced at 4 h and 24 h in the sensitive OVCAR-8 cells while it was not affected in NCI/ADR-RES cells. DNA repair mediated by *RAD51* is both p53-dependent and -independent in cells. This would suggest that the repair of DSB’s induced by NCX4040 is p53-independent. Furthermore, DSB’s formed in OVCAR-8 cells are repaired while DSB’s in NCI/ADR-RES cells are not repaired or repaired at a much slower rate. *RAD51* was significantly decreased in both cells at 48 h when cells were undergoing cellular death. Growth arrest and DNA damage inducible protein 45 gene (*GADD45*), involved in demethylation of DNA [45] as well as cell cycle arrest, and apoptosis [46], was also markedly elevated in OVCAR-8 cells at all time points. In contrast, *GADD45* was decreased in NCI/ADR-RES cells following NCX4040 treatment, again suggesting decreased DNA repair activities in these cells. Induction of *GADD45* has been previously reported in OVCAR-8 cells as well as other ovarian cell lines without p53 [41]. *OGG1*, a gene responsible for carrying out repair of oxidative damage (8-oxoguanine), was slightly decreased at 4 h in OVCAR-8 cells, but remained unchanged in NCI/ADR-RES cells at 4, 24 and 48 h. Our recent studies using microarray analysis have shown that *SOD2* was significantly (2–3-fold) induced by NCX4040 treatment in OVCAR-8 cells. In addition, we found that *NOX4*, a NADPH oxidase responsible for generating O_2_^●−^, was significantly induced in both cell lines (manuscript in preparation).

These studies clearly support our findings that NCX4040 induces significant oxidative stress in both cells due to rapid formation of reactive species which deplete GSH and induce DSB’s; however, decreased amounts of reactive species were detected in NCI/ADR-RES cells due to higher expressions of detoxification enzymes. Because of this rapid detoxification, decreased DNA damage was also observed in NCI/ADR-RES cells. DNA damage appeared to be repaired in OVCAR-8 cells but poorly or not repaired in NCI/ADR-RES cells. 

It should be noted that significantly more GSH was depleted in NCI/ADR-RES cells at higher concentrations of NCX4040 and this may result from the enhanced detoxification of hydrogen peroxide by GSH/GPx as well as increased metabolism of NCX4040 by GSH/GST systems. GSH/GST has been reported to be involved in the biotransformation of NCX4040 [31]. At these higher concentrations of NCX4040, significantly more apoptotic cells were also detected in NCI/ADR-RES cells than in OVCAR-8 cells, suggesting more than one mechanism of tumor cell death is involved at higher drug concentrations of NCX4040. At physiological and lower doses, ROS/RNS play significant roles in cell killing.Resistance observed to NCX4040 in NCI/ADR-RES cells may result from this increased detoxification of these species, decreasing DNA damage and cell death. At much higher NCX4040 concentrations, various mechanisms of cell killing may be involved e.g., formation of a quinone-methide in the killing of tumor cells by NCX4040. However, we believe that the formation of the quinone-methide and subsequent conjugation with GSH, catalyzed by GSH/GST system, is a detoxification of NCX4040 and not an activation for the cell killing. It is, however, possible that at these high NCX4040 concentrations cellular defense systems are overwhelmed/compromised and the resulting quinone-methide could compete and induce cellular damage, e.g., alkylating DNA and cellular proteins, causing cell death. 

Because FETPPS, a scavenger of peroxynitrite, inhibited NCX4040-dependent cytotoxicity, it is reasonable to suggest that NCX4040 formed ^●^NO which then reacted with superoxide to generate peroxynitrite that may be responsible for cell death in these tumor cells as proposed in Figure 10. 

Based these findings we have now designed various analogs of NCX4040 for synthesis and further studies. It is anticipated that these analogs will enhance the formation of ROS and peroxynitrite while concomitantly decreasing the formation of quinone methide in tumor cells. It is anticipated that these analogs will not only further delineate mechanism of actions of NCX4040 but will also provide a more active anticancer agent for the treatment of cancers in the clinic.

While immunological tumor cell death with NCX4040 is not known at this time, our recent results using microarray analysis with low doses of NCX4040 indicate that NCX4040 significantly induces *IL-6* gene and other immune responses genes (NFkB, TNF, etc.) in OVCAR-8 and NCI/ADR-RES cells. In addition, we have found significant modulation of cyclooxygenase-2 (*COX-2*) by NCX4040 in both cell lines. Modulation of *COX-2* has been implicated in the mechanism of actions of NCX4040 [47,48,49,50]. Oxidative stress and inflammation have been reported be closely associated with cancer and apoptosis [51]. Work is in progress to elucidate the roles of *COX2*, immune response genes and oxidative stress in the tumor cell death in ovarian tumor cells.

## 4. Materials and Methods

The nitric oxide donor, 2-(acetyl)benzoic acid 4-(nitroxymethyl) phenylester (NCX4040) and N-acetylcysteine (NAC) were obtained from Sigma Chemicals (St. Louis, MO, USA). Stock solutions of NCX4040 were prepared in DMSO and stored at −80 °C. Fresh drug solutions prepared from stock solutions were used in all experiments. Topotecan.HCl (TPT) and FeTPPS (5,10,15,20 tetrakis(4-sulfonatophenyl)propyporphirinato iron (III), chloride were purchased from Cayman Chemicals (Ann Arbor, MI, USA). Stock solution of TPT was prepared in double distilled water; in some cases, TPT was dissolved in DMSO, and stored at −80 °C. NAC was dissolved in DMSO and stored at −80 °C. FeTPPS was dissolved in sterile PBS prior to use.

### 4.1. Cell Culture

Authenticated human ovarian OVCAR-8 and ADR-selected OVCAR-8 cells (NCI/ADR-RES) cells were obtained from the NCI-Frederick Cancer Center (Frederick, MD, USA). Cells were cultured in Phenol Red-free 1640 RPMI media containing 10% fetal bovine serum and antibiotics (complete media). Cells were routinely used for 20–25 passages, after which the cells were discarded, and a new cell culture was started from fresh, frozen stock. 

### 4.2. Cytotoxicity of NCX4040 and Topotecan in Tumor Cells 

The cytotoxicity studies were carried out with a cell growth inhibition assay and Trypan Exclusion methods [5]. Briefly, 100,000–150,000 cells/well were seeded onto a 6-well plate (in duplicate) and allowed to attach for 18 h. Various concentrations of drugs (NCX4040 or TPT) were added to cells (OVACAR-8, NCI/ADR-RES) in fresh complete media (2 mL). DMSO (0.01–0.1%) was included as the vehicle control when used. Following trypsinizing, surviving cells were collected and counted in a cell counter (Beckman, Brea, CA, USA). For the trypan blue exclusion assay 15 μL of cell mixtures were combined with 15 µL of trypan blue and counted in a T20 automatic cell counter (Bio-Rad, Hercules, CA, USA). 

### 4.3. Flow Cytometric Analysis of Intracellular Glutathione

Intracellular glutathione was determined by adding monochlorobimane dye (mBCl, 10 µM final; Life Technologies, Carlsbad, CA, USA) to each sample for 15 min at 37 °C, 5% CO_2_ atmosphere prior to examination. Propidium iodide (PI) was added (final concentration of 5 µg/mL) to samples before flow cytometric analysis using a LSRFortessa flow cytometer (Benton Dickinson, San Jose, CA, USA) equipped with FACSDiVa software. mBCl and PI was excited using a 405 nm and 561 nm laser and detected using a 530/30 nm and 582/15 nm filter, respectively. For each sample, 10,000 cells were analyzed using FACSDiVa software.

### 4.4. Flow Cytometric Analysis of Mitochondrial ROS 

Analysis of mitochondrial ROS was determined by loading the cells with MitoSox Red (5 µM final; Life Technologies) for 30 min at 37 °C, 7% CO_2_ atmosphere before the addition of the drug. Cells were examined at 1 h intervals with the addition of Sytox Blue as a vital dye by flow cytometry. A LSRFortessa flow cytometer (Benton Dickinson, San Jose, CA, USA), equipped with FACSDiVa software, was used to analyze all samples. MitoSox and Sytox Blue were excited using a 561 nm and 405 nm laser and detected using a 610/20 nm and 450/50 nm filter, respectively. For each sample, 10,000 cells were analyzed using FACSDiVa software.

### 4.5. H2AX Phosphorylation Assay

H2A.X phosphorylation assay kit (Catalog # 17-344, Upstate, NY, USA) was used to detect H2A.X phosphorylation according to the manufacturer’s instructions. Briefly, cell samples were collected following drug treatment (60 min) and washed with PBS. Samples were resuspended in 1X Fixation solution (50 µL) for 20 min on ice, washed in 1X PBS (2×) and then resuspended in 1X Permeabilization solution (50 µL). 3.5 µL of either anti-phospho-Histone H2A.X (Ser139) FITC conjugate or negative control mouse IgG-FITC conjugate were added to cell samples. Following incubation on ice for 20 min, 1X wash solution (100 µL) was added to each sample, pelleted, and resuspended in 150 µL of 1X PBS. Samples were analyzed using a BD LSR II flow cytometer equipped with FACSDiVa software. Only single cells, devoid of debris, were used in the analysis.

### 4.6. Flow Cytometric Analysis of Caspase Activity

Caspase activity for caspase-3/7-like enzymes was determined using a CaspaTag in situ assay kit (Chemicon, Billerica, MA, USA) according to the manufacturer’s instructions. Briefly, 1 h prior to analysis, 10 µL of a 30× CaspaTag reagent stock solution was added to 300 µL of cells and incubated for 1 h, washed in 2 mL of CaspaTag wash buffer, and then resuspended in 500 µL of 1X PBS. Propidium iodide (PI) was added (final concentration of 5 µg/mL) to samples prior to flow cytometric analysis. A LSRFortessa flow cytometer (Benton Dickinson, San Jose, CA, USA) equipped with FACSDiVa software was utilized for analysis of cells. CaspaTag and PI were excited using a 488 nm and 561 nm laser and detected using a 530/30 nm and 582/15 nm filter, respectively. A total of 10,000 cells (from each sample) were analyzed using FACSDiVa software.

### 4.7. Flow Cytometric Analysis of Annexin-V Binding

Changes in membrane phosphatidylserine symmetry were determined using Annexin-5 V binding assay kit (Trevigen, Gaithersburg, MD, USA) according to the manufacturer’s instructions. Briefly, cells were washed in 1X PBS, then incubated with 1 µL Annexin-V FITC and propidium iodide (PI) in Annexin-V binding buffer for 15 min at room temperature. After this time, the samples were diluted with 1X binding buffer and examined immediately by flow cytometry. Cells were analyzed using a LSRFortessa flow cytometer (Benton Dickinson, San Jose, CA, USA) equipped with FACSDiVa software. Annexin-V FITC and PI were excited using a 488 nm and 561 nm laser and detected using a 530/30 nm and 582/15 nm filter, respectively. For each sample, 10,000 cells were analyzed using FACSDiVa software.

### 4.8. Real-Time RT-PCR

The expression levels of selected transcripts were confirmed by real-time polymerase chain reaction (RT-PCR) using absolute SYBR green ROX Mix (ThermoFisher Scientific, Rochester, NY, USA) as previously described [52]. OVCAR-8 and NCI/ADR-RES cells were treated with NCX4040 (5 µM) for 4, 24, 48 h in the complete media. Following treatment, cells were washed once with PBS (pH 7.4) and RNA was extracted with TRIzol (Ambion Life Technology, Grand Island, NY, USA). RNA was isolated and purified with RNA easy mini kit columns (Qiagen, Valencia, CA, USA). Data were analyzed using ΔΔCt method of relative quantification in which cycle times were normalized to β-actin (or GADPH) from the same sample. Primers for the selected genes were designed using Primer Express 1.0 software and synthesized by Integrated DNA Technologies, San Diego, CA, USA) and in some cases were purchased from Origene Technologies (Gaithersburg, MD, USA). All real-time fluorescence detection was carried out on an iCycler (Bio-Rad, Hercules, CA, USA). 

### 4.9. Statistical Analysis

The results are expressed as mean ± SEM of minimum of 3 experiments (n = 3). One-way analysis of variance (ANOVA) was used for statistical analysis using Graph Pad Prism (GraphPad Software, Inc., La Jolla, CA, USA). For multiple comparisons, the Tukey multiple comparison’s test was utilized; results were considered statistically significant when *p* < 0.05.

## 5. Conclusions

Our studies show that NCX4040 treatment resulted in the formation of significant amounts of ROS/RNS and DSB’s in OVCAR-8 cells, leading to cell death. In contrast, NCI/ADR-RES cells which are resistant to lower concentrations of NCX4040 formed lower amounts of ROS and ROS/RNS-induced DSB’s. These observations indicate that ROS/RNS plays a significant role in NCX4040 cytotoxicity. Furthermore, our studies show that at much higher concentrations of NCX4040, more GSH was depleted as well as more apoptotic cells were detected in NCI/ADR-RES cells than in OVCAR-8 cells. These results would then suggest that at higher concentrations of NCX4040, many other mechanisms may also be operative in NCX4040-induced tumor cell death, in addition to GSH depletion and oxidative stress induced by both ROS/RNS and quinone methide formation. 

## Figures and Tables

**Figure 1 ijms-23-08611-f001:**
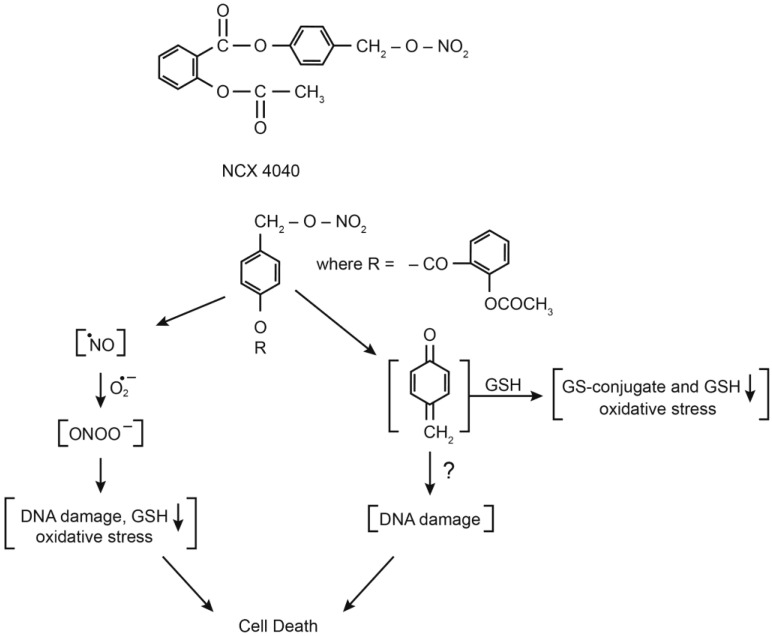
Structure of NCX4040 and its known/proposed mechanisms of cytotoxicity.

**Figure 2 ijms-23-08611-f002:**
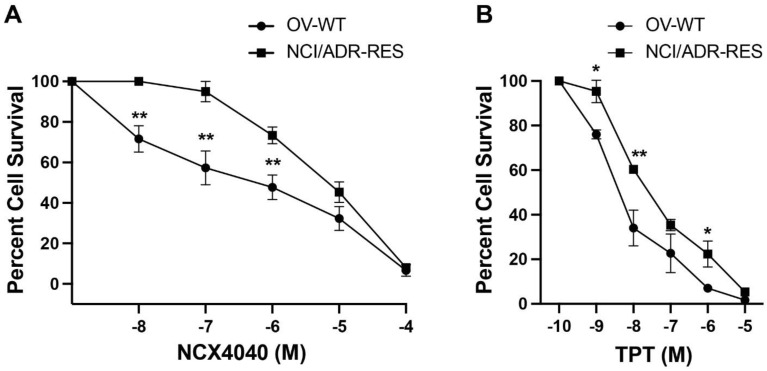
Cytotoxicity of NCX4040 in (**A**) and topotecan (TPT, (**B**)) in WT OVCAR-8 and NCI/ADR-RES cells at 48 h drug exposures, respectively. * *p* and ** *p*-values, <0.05 and <0.005, respectively, compared to concentration-matched treatment.

**Figure 3 ijms-23-08611-f003:**
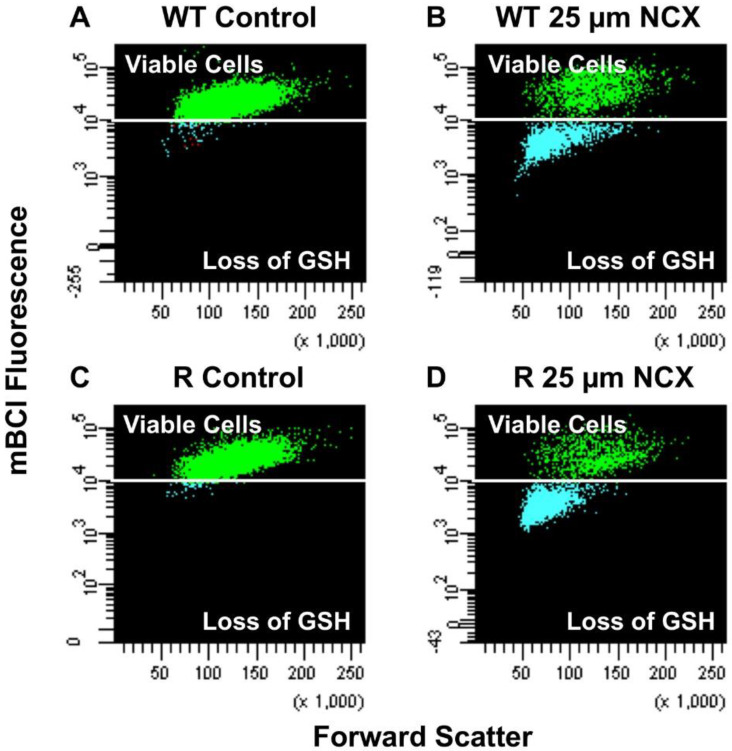
Percent GSH depletion in OVCAR-8 (**A**,**B**) and in NCI/ADR-RES cells (**C**,**D**) and concentration dependence of GSH depletion by NCX4040 for 4 h (**E**). Representative plots (**A**–**D**) are shown here. *** *p*-values, <0.001, compared to concentration-matched NCI/ADR-RES.

**Figure 4 ijms-23-08611-f004:**
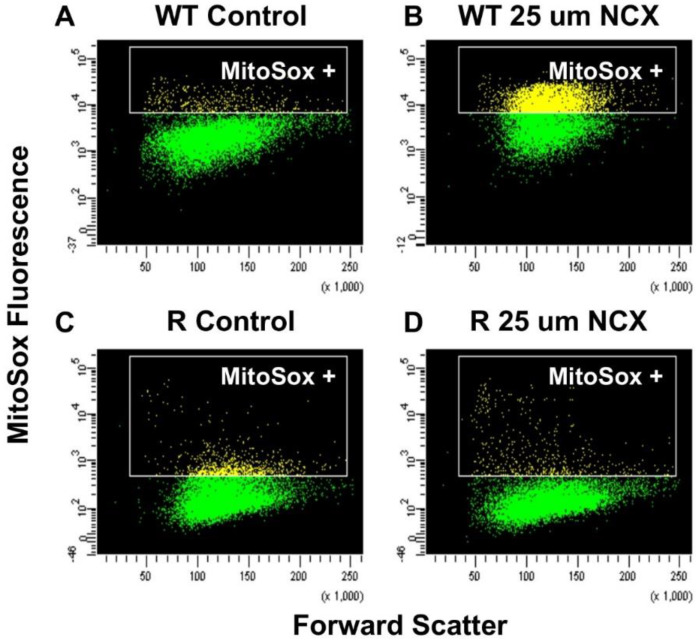
Formation of Mitosox positive cells in OVCAR-8 and NCI/ADR-RES cells during incubation with NCX4040 (25 µM) with time (**E**). Plots (**A**–**D**) were obtained at 2 h of drug treatment and representative plots are shown here. *** *p*-values <0.001 compared to time-matched control and NCI/ADR-RES.

**Figure 5 ijms-23-08611-f005:**
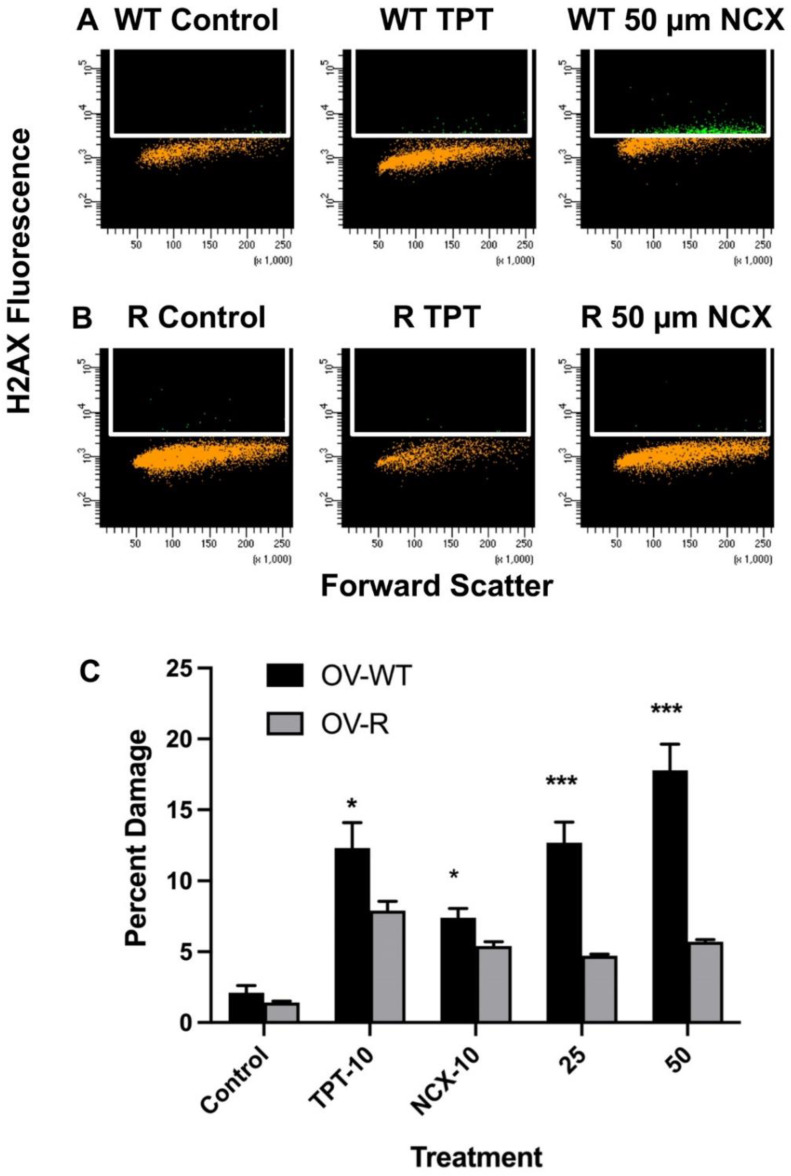
Detection of DNA-double strand breaks produced following treatment with different concentrations of NCX4040 in OVCAR-8 and NCI/ADR-RES cells for 60 min and with different concentrations (**C**). TPT (10 µM) was used as a control. Representative plots (**A**,**B**) are shown here. * and *** *p*-values, <0.05, and <0.001, respectively, compared to concentration-matched treatment.

**Figure 6 ijms-23-08611-f006:**
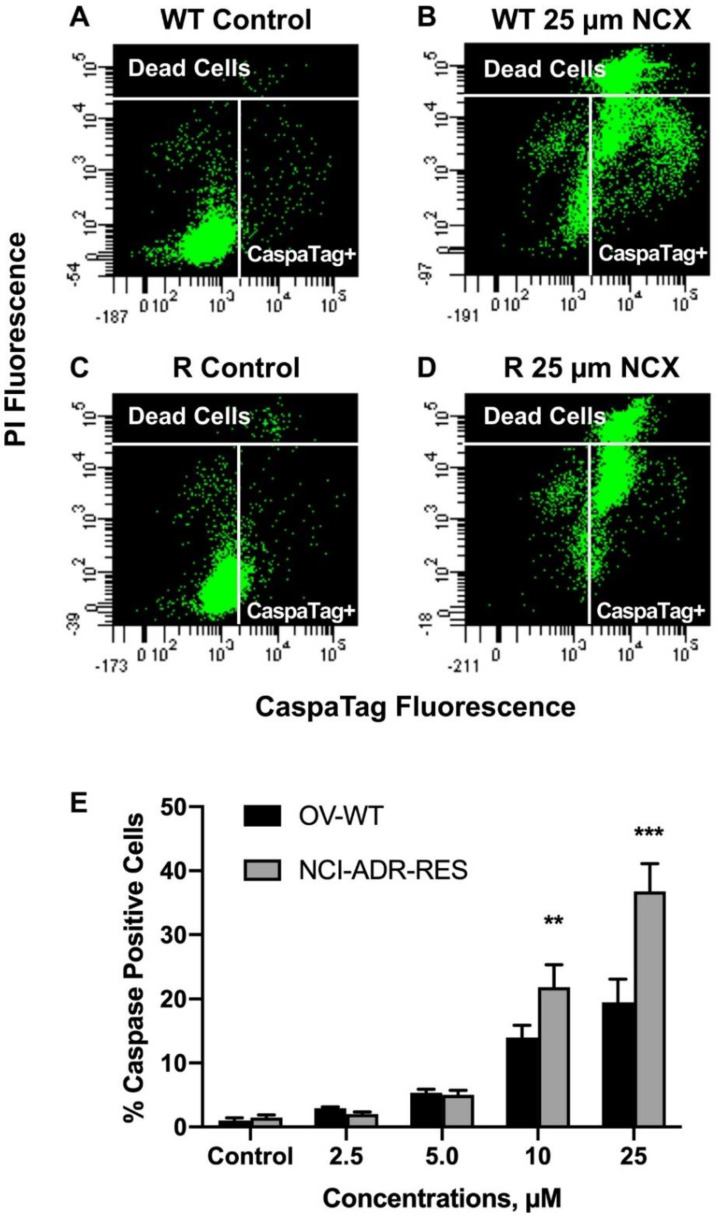
Detection of CaspaTag positive cells (**A**,**B**) OVCAR-8 and (**C**,**D**) in NCI/ADR-RES cells and with different concentrations of NCX4040 for 24 h (**E**). Representative plots (**A**–**D**) are shown here. ** and *** *p*-values, <0.005, and <0.001, respectively, compared to concentration-matched treatment.

**Figure 7 ijms-23-08611-f007:**
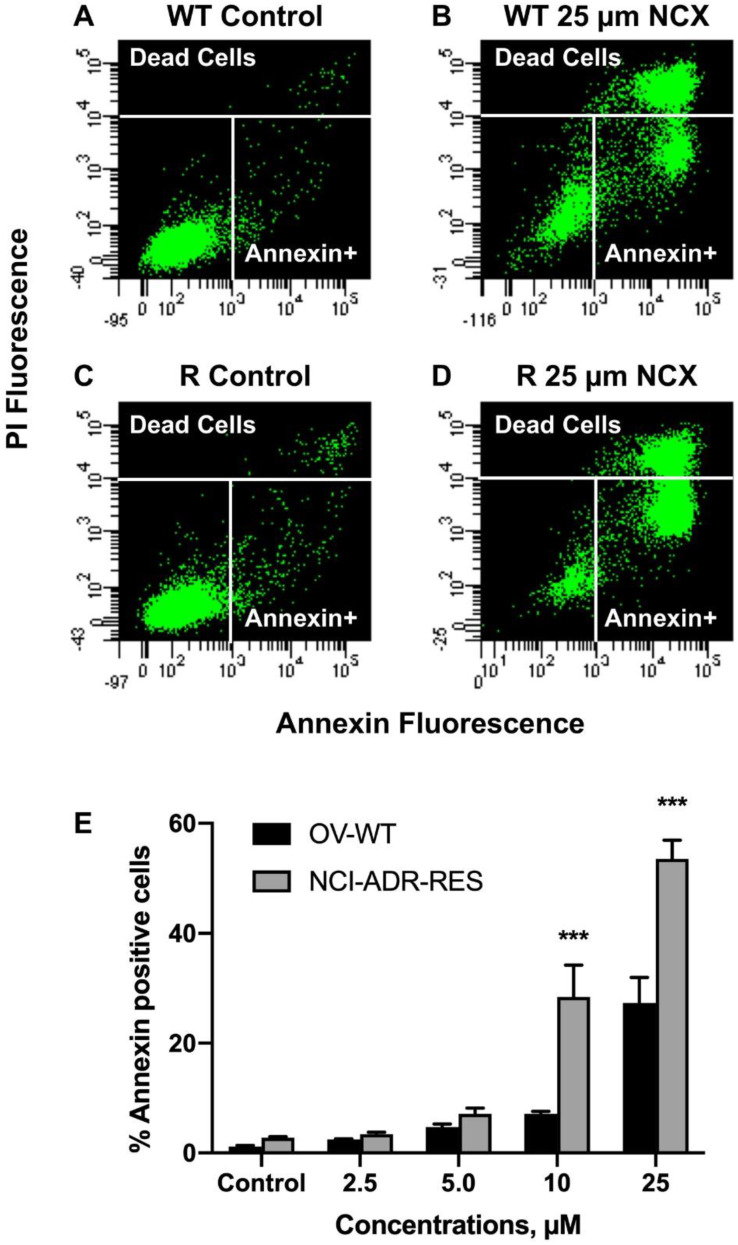
Detection of Annexin positive cells (**A**,**B**) OVCAR-8, and (**C**,**D**) in NCI/ADR-RES cells and with different concentrations of NCX4040 for 24 h (**E**). Representative plots (**A**–**D**) are shown here. *** *p*-values, <0.001, compared to concentration-matched treatment.

**Figure 8 ijms-23-08611-f008:**
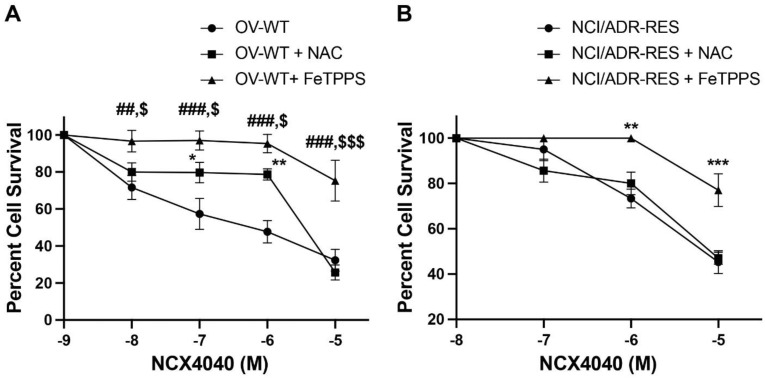
Effects of NAC (2 mM) and FeTPPS (5 µM) on the cytotoxicity of NCX4040 in OVCAR-8 (**A**) and NCI/ADR-RES (**B**) cells following 48 h exposure. NAC or FeTPPS were preincubated with cells for 20–30 min before adding NCX4040. *, ** and *** *p*-values, <0.05, <0.005 and <0.001, respectively, compared to concentration-matched drug alone treatment. ##, ### *p*-values, <0.005 and <0.001, respectively, compared to concentration-matched drug alone treatment. $, $$$ *p*-values, <0.05, and <0.001, respectively, compared to concentration-matched drug + NAC treatment.

**Figure 9 ijms-23-08611-f009:**
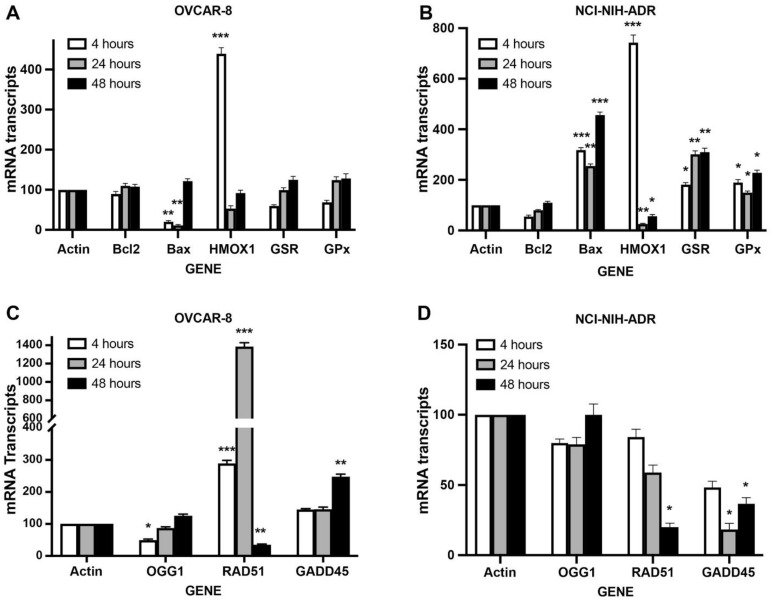
RT-PCR analysis of apoptosis- and oxidative stress-related genes differentially expressed by NCX4040 (5 µM) in OVCAR-8 (**A**) and NCI/ADR-RES (**B**) cells and DNA damage and DNA repair genes in OVCAR-8 (**C**) and NCI/ADR-RES (**D**) following 4, 24 and 48 h NCX4040 treatment. RT-PCR was carried out as described in the methods section. *, ** and *** *p*-values, <0.05, <0.005 and <0.001, respectively, compared to actin controls.

**Figure 10 ijms-23-08611-f010:**
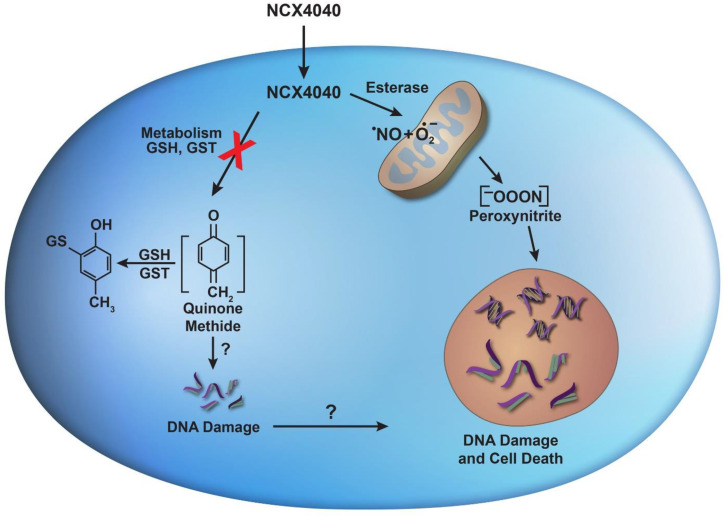
Proposed formation of reactive free radical species, ^●^NO, O_2_^●−^, and O=N-O-O^−^ following cellular metabolism of NCX4040 in tumor cells and role of peroxynitrite in tumor cell death.

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
