# Peer review of "Molecular Mechanisms of Cytotoxicity of NCX4040, the Non-Steroidal Anti-Inflammatory NO-Donor, in Human Ovarian Cancer Cells"

_ijms, 2022, doi:10.3390/ijms23158611_

Round 1

Reviewer 1 Report

The work "Role of Free Radicals and Peroxynitrite in Mechanisms of Cyto- 2
toxicty of Ncx4040 in Human Ovarian Cancer Cells" by Birandra Sinha and coworkers is an interesting study on the cytotoxicity of NCX4040 compound, "the non-steroidal anti-inflammatory-NO donor". I recommend publication after the important following changes:

1. Title needs to be changed. The role of free radicals and peroxynitrite in the cytotoxicity of Ncx4040 has not been clearly confirmed. The results indicate the induction of oxidative stress and the participation of biological oxidants of radical (O2*-, NO2*, CO3*-) or non-radical nature (H2O2, ROOH, ONOO-).

2. line 34: "Activities of ●NO in cells have been shown to result from formation of reactive metabolites, NO+, N2O3, and •_OONO." NO+ formation in cells is very unlikely. A species like *-OONO does not exist.

3. line 47: "NCX4040 is a nitro derivative of aspirin and has been shown to release ●NO following hydrolysis by tumor esterases". General remark: Organic nitrates are called NO donors, although to my knowledge their mechanism of action has not been uniquely defined and it is not clearly shown that they directly release NO.

4. Figure 1 - In Figure 1, there are several wrongly drawn structures - for example: NOOOH, quinone methide...

5. Figure 4. How to explain the difference of Mitosox signal between "OV-WT +25 µM NCX" and "OV-R + 25 µM NCX"? This figure suggests that only in the case of OV-WT, incubation with NCX4040 efficiently generated the oxidative stress. How does Mitosox signal depend on NCX4040 concentration and incubation time?

6. line 170: "reactive peroxynitrite" - what does it mean?

7. General remark: the article does not directly show the participation of free radicals (the participation of any free radical has not been confirmed, for example, with the use of spin traps and EPR) in the cytotoxicity of NCX4040. Similarly, the participation of peroxynitrite in the cytotoxicity mechanism of NCX4040 was also not shown (e.g. with the use of boronate probes) - the discussion and the general conclusion should be modified.

Author Response

  1. Title needs to be changed.The role of free radicals and peroxynitrite in the cytotoxicity of Ncx4040 has not been clearly confirmed. The results indicate the induction of oxidative stress and the participation of biological oxidants of radical (O2*-, NO2*, CO3*-) or non-radical nature (H2O2, ROOH, ONOO-).

I agree with the reviewer that the role of free radicals in cytotoxicity is not established here, we have now changed the title to “ Molecular mechanisms of cytotoxicity of NCX4040, the non-steroidal anti-inflammatory-NO donor, in human ovarian cancer cells”

  1. line 34: "Activities of ●NO in cells have been shown to result from formation of reactive metabolites, NO+, N2O3, and •_OONO." NO+ formation in cells is very unlikely.A species like *-OONO does not exist.

I agree with the reviewer that ●-OONON does not exit. Has been corrected. However, various researchers have suggested that NO+ is formed in cells. Here is a recent paper published on this subject showing the formation of NO+ in cells

Anatoly F. Vanin , Applied Magnetic Resonance volume 51, pages 851–876 (2020)

  1. line 47: "NCX4040 is a nitro derivative of aspirin and has been shown to release ●NO following hydrolysis by tumor esterases". General remark: Organic nitrates are called NO donors, although to my knowledge their mechanism of action has not been uniquely defined and it is not clearly shown that they directly release NO.

I agree with the reviewer that the mechanism is not completely understood, however, NCX4040 does release ●NO as shown by EPR by Kuppusamy’s group (reference-10).

  1. Figure 1 - In Figure 1, there are several wrongly drawn structures - for example: NOOOH, quinone methide...

These have been corrected.

  1. Figure 4. How to explain the difference of Mitosox signal between "OV-WT +25 µM NCX" and "OV-R + 25 µM NCX"? This figure suggests that only in the case of OV-WT, incubation with NCX4040 efficiently generated the oxidative stress. How does Mitosox signal depend on NCX4040 concentration and incubation time?

It is correct that only WT OV cells form significant amounts of ROS (or Mitosox signals). Our preliminary experiments showed that with higher concentrations of NCX (e.g., 25 µM NCX), there were significant differences between two cell lines. Therefore, we focused on this concentration and not on lower concentrations of NCX. More Mitosox signals are detected with time as shown in the Figure-4.

  1. line 170: "reactive peroxynitrite" - what does it mean?

Because peroxynitrite reacts with lipids, DNA, and proteins, I used the term reactive.

  1. General remark: the article does not directly show the participation of free radicals (the participation of any free radical has not been confirmed, for example, with the use of spin traps and EPR) in the cytotoxicity of NCX4040.Similarly, the participation of peroxynitrite in the cytotoxicity mechanism of NCX4040 was also not shown (e.g. with the use of boronate probes) - the discussion and the general conclusion should be modified.

I agree with this reviewer that the formation of free radical species was not directly observed by spin trapping ESR methods. While we did not use boronate probes we utilized FeTPPS to confirm the participation of peroxynitrite in the cytotoxicity of NCX4040. It should be noted that FeTPPS is  a very specific inhibitor of peroxynitrite and has been extensively used by various investigators to decipher the roles of peroxynitrite in cells. In our studies, we found that it completely attenuated cytotoxicity of NCX4040 in both cell lines, indicating that peroxynitrite was involved in NXC4040 cytotoxicity.

Reviewer 2 Report

In the present article the authors identify the mechanism of action of the NO-releasing drug NCX4040 in ovarian cells. They have identified that the action of NCX4040 is mediated by ROS or RNS. Specifically, the found that in OVCAR-3 cells NCX4040 acts through ROS while in NCI/ADR-RES (adriamycin resistant) cells the main mechanism is RNS. This action was completely inhibited by the FeTPPS peroxynitrite scavenger. The study is interested with a significant degree of novelty and well-designed.

I only have one comment/suggestion.

It would have been interesting if the authors would have determined the expression levels of SOD1, catalase and glutathione peroxidase, transferase in these cell lines. This experiment would support their implication in generation of more free radicals in NCI/ADR-RES compared to OVCAR-3. The authors have RNA from these cells and would be an easy experiment to do.

Author Response

We have done all these genes. Our microarray analysis indicated that SOD2 was induced in only WT OV cells at 4 and 48 h. Catalase was not induced in either cell lines. Glutathione peroxidase was not induced in WT-OV cells; however, it was significantly induced in NCI/ADR-RES cells as shown in by RT-PCR (Figure-9). Interestingly, we found that NOX4 was significantly induced in both cell lines. We have now included this in our revised manuscript (page-30, 1st paragraph)

We have done all these genes. Our microarray analysis indicated that SOD2 was induced in only WT OV cells at 4 and 48 h. Catalase was not induced in either cell lines. Glutathione peroxidase was not induced in WT-OV cells; however, it was significantly induced in NCI/ADR-RES cells as shown in by RT-PCR (Figure-9). Interestingly, we found that NOX4 was significantly induced in both cell lines. We have now included this in our revised manuscript (page-30, 1st paragraph)

We have done all these genes. Our microarray analysis indicated that SOD2 was induced in only WT OV cells at 4 and 48 h. Catalase was not induced in either cell lines. Glutathione peroxidase was not induced in WT-OV cells; however, it was significantly induced in NCI/ADR-RES cells as shown in by RT-PCR (Figure-9). Interestingly, we found that NOX4 was significantly induced in both cell lines. We have now included this in our revised manuscript (page-30, 1st paragraph)

Round 2

Reviewer 1 Report

1.     1. “I agree with the reviewer that ●-OONON does not exit. Has been corrected. However, various researchers have suggested that NO+ is formed in cells. Here is a recent paper published on this subject showing the formation of NO+ in cells Anatoly F. Vanin , Applied Magnetic Resonance volume 51, pages 851–876 (2020)”

Some nitric oxide metabolites or forms are sometimes called NO+ carriers e.g. DNIC or RSNO, however, NO+ itself is a strong oxidant and a very reactive electrophile. First: there are practically no oxidants in biological systems capable of oxidizing NO to NO+. Secondly - even if NO+ was formed, it would undergo very fast hydrolysis in the aqueous solution to NO2- via H2NO2+ and HNO2. Please, read: Martin N. Hughes “Relationships between nitric oxide, nitroxyl ion, nitrosonium cation and peroxynitrite” Biochimica et Biophysica Acta 1411 (1999) 263^272. The formation of “free” NO+ in vivo is a scientific fantasy.

2.      “Because peroxynitrite reacts with lipids, DNA, and proteins, I used the term reactive.”

Please, just use term peroxynitrite.

3.      General remark: the article does not directly show the participation of free radicals (the participation of any free radical has not been confirmed, for example, with the use of spin traps and EPR) in the cytotoxicity of NCX4040.Similarly, the participation of peroxynitrite in the cytotoxicity mechanism of NCX4040 was also not shown (e.g. with the use of boronate probes) - the discussion and the general conclusion should be modified.

“I agree with this reviewer that the formation of free radical species was not directly observed by spin trapping ESR methods. While we did not use boronate probes we utilized FeTPPS to confirm the participation of peroxynitrite in the cytotoxicity of NCX4040. It should be noted that FeTPPS is  a very specific inhibitor of peroxynitrite and has been extensively used by various investigators to decipher the roles of peroxynitrite in cells. In our studies, we found that it completely attenuated cytotoxicity of NCX4040 in both cell lines, indicating that peroxynitrite was involved in NXC4040 cytotoxicity.”

The use of FeTPPS is not enough to confirm the participation of peroxynitrite in the mechanism of NCX4040 cytotoxicity. The mechanism of action of FeTPPS in cells is not well established. It can be also reactive towards H2O2 or superoxide. The formation of peroxynitrite is a hypothesis.

Author Response

The conclusions have been completely modified as suggested by this reviewer

Reviewer 2 Report

The authors have responded to all comments and the manuscript in my opinion can now be found accepted for publication.

Author Response

Reviewer had no further questions.